# Surgical Site Cytology to Diagnose Spinal Lesions

**DOI:** 10.3390/diagnostics12020310

**Published:** 2022-01-26

**Authors:** Leon-Gordian Koepke, Annika Heuer, Martin Stangenberg, Marc Dreimann, Lutz Welker, Carsten Bokemeyer, André Strahl, Anne Marie Asemissen, Lennart Viezens

**Affiliations:** 1Department of Trauma and Orthopedic Surgery, Division of Spine Surgery, University Medical Center Hamburg Eppendorf, Martinistraße 52, D-20246 Hamburg, Germany; ann.heuer@uke.de (A.H.); m.stangenberg@uke.de (M.S.); m.dreimann@uke.de (M.D.); l.viezens@uke.de (L.V.); 2Institute of Pathology with the Sections Molecular Pathology and Cytopathology, University Medical Center Hamburg Eppendorf, Martinistraße 52, D-20246 Hamburg, Germany; l.welker@uke.de; 3Department of Oncology, Hematology and Bone Marrow Transplantation with Section Pneumology, Hubertus Wald University Cancer Center, University Medical Center Hamburg Eppendorf, Martinistraße 52, D-20246 Hamburg, Germany; c.bokemeyer@uke.de (C.B.); a.asemissen@uke.de (A.M.A.); 4Department of Trauma and Orthopedic Surgery, Division of Orthopedics, University Medical Center Hamburg Eppendorf, Martinistraße 52, D-20246 Hamburg, Germany; a.strahl@uke.de

**Keywords:** spinal neoplasms, biopsy, bone marrow, spine, cytological technique, standard of care, metastases

## Abstract

Patients with new-onset malignant spinal lesions often have an urgent need for local spine intervention and systemic therapy. For optimal management, it is crucial to diagnose the underlying disease as quickly and reliably as possible. The aim of our current study was to determine the feasibility, sensitivity, specificity, and diagnostic certainty of complementary cytological evaluation of spinal lesions suspected of malignancy. In 44 patients, we performed histopathological biopsies and in parallel cytologic preparations from the malignant site. Cytological smears were prepared and stained for May-Grunwald and Giemsa. Bone biopsies were histopathologically analyzed according to the existing standard-of-care practices. In 42 of 44 cases (95%), a cytological sample was successfully obtained. In 40 cases (95.2%, Cohen’s kappa: 0.77), the cytological diagnosis agreed with the histological diagnosis regarding the identification of a malignant lesion. This resulted in a sensitivity of 97% and a specificity of 80% as well as a diagnostic safety of 95%. Cytological analysis in the context of spinal surgery proved sufficient to establish a diagnosis of malignancy or its exclusion, expanding the existing diagnostic spectrum. Furthermore, implementation of this process as a routine clinical diagnostic might shorten the time to diagnosis and improve the treatment of this vulnerable patient group.

## 1. Introduction

Spinal neoplasms can be roughly divided into three groups. Spinal manifestations of a primary tumor located in another organ system are by far the largest group. Within this group, solid tumors can be distinguished from hematologic tumors. This group is followed by benign tumors. The lowest probability of occurrence is found in the group of primary malignant spinal tumors [1].

More than 70% of patients with a malignant primary tumor develop metastases in the spine during the disease [2], and in up to 30% of patients, spinal complaints due to metastases are the initial symptom of the tumor disease [3]. Thus, it is common for patients with previously undiagnosed malignancies to present to the clinic with spinal manifestations of distant malignant tumors. When patients present with malignant spinal lesions, urgent or emergency surgical treatment of the spine is indicated in cases with immobilizing pain, unstable fractures, or compression of the spinal cord [4]. Furthermore, prompt diagnostic confirmation of the underlying disease to initiate oncologic-specific therapy is paramount in patients who are often critically ill [5]. The gold standard for confirming the diagnosis of the disease is histopathological examination with additional immunohistochemistry (IHC) [6]. In contrast to soft tissue biopsies, material from osseous sites needs to undergo decalcification for several days before being embedded in paraffin wax, sliced, stained, and analyzed.

Generally, in addition to histopathological examination of specimens, cytological specimen evaluation is possible. In comparison to classical histology, in which cellular tissue composites are evaluated two-dimensionally, cytology is used to evaluate individual cells, cell fragments, or cell clusters [7,8]. As a result, cytological procedures have few requirements in terms of the type and quantity of material. Furthermore, cytological laboratories are inexpensive and can be operated independently of pathological institutes [7]. In addition, cytological evaluation of sample material, including IHC, genetic analysis, and flow cytometry, can be performed quickly, since the material does not have to be processed in a time-consuming manner before staining and evaluation.

For example, in the diagnosis of oral, urogenital, gynecological, or bronchopulmonary malignancies, cytology is already clinically relevant [8,9,10,11,12,13,14]. Studies showed that cytological evaluation of bronchopulmonary lesions suspected of malignancy has a sensitivity of 94.8% and a specificity of 98.8% [8]. Overall, the addition of cytologic diagnosis to histologic diagnosis has expanded the diagnostic spectrum in the diagnosis of malignant neoplasms and increased diagnostic certainty. Cytological evaluation of spinal tumors by fine-needle aspiration biopsy (FNAB) has also been extensively studied in the literature [15,16,17]. However, in general, cytological evaluation of spinal lesions is performed in a radiologically guided manner independent of spinal surgery and in the elective program. To date, no data have been published regarding the diagnostic value of intraoperative cytologic evaluation of spinal lesions in the emergency or highly urgent spinal surgery settings.

The aim of our study was to conduct the first investigation into the intraoperative cytological evaluation of spinal lesions suspected of malignancy regarding patient safety, feasibility, sensitivity, specificity, diagnostic certainty, and, thus, the expansion of the diagnostic spectrum in a patient collective with the need for emergency or highly urgent spinal surgery.

## 2. Materials and Methods

This study is reported according to the guideline for Strengthening The Reporting of Observational Studies in Epidemiology (STROBE) [18]. All methods were conducted in accordance with relevant guidelines and regulations. This study and its protocols were approved by the ethics committee of the Hamburg Medical Association. The patient data were anonymized; therefore, according to the Hamburg ethics guidelines, informed consent was not needed (ethics vote: 2021-300102-WF).

Between February and November 2021, 44 patients with a new-onset spinal manifestation of tumor disease underwent spinal surgery at our institution. Specimens were obtained for histopathological and cytological evaluation during spinal surgery according to current in-house clinical standards.

For this purpose, classical tissue samples were collected during spine surgery to perform histology. In addition, the cytological samples were obtained by spinal bone marrow aspiration (SBMA) as follows: The usually transpedicular puncture of the vertebral body with the lesion suspected of malignancy was performed using a Jamshidi biopsy needle. The core was then removed from the needle. The hollow needle was then used to aspirate 3–5 mL of bone marrow into 2 mL of citrate using a syringe, with a sharp pull. For additional analysis with flow cytometry and fluorescence in situ hybridization (FiSH), two other 10 mL syringes, both loaded with 1 mL EDTA, were filled with 3 mL aspirate; for cytogenetic analysis, the sterile syringe was loaded with 1 mL heparin. After aspiration, the syringe was briefly and gently swirled to mix the bone marrow with the citrate. The sample was then sent for processing. Then, the Jamshidi biopsy needle was advanced through the vertebral lesion without a core to obtain the core needle biopsy for histological evaluation.

Histological and cytological specimens were evaluated simultaneously and independently. Histological diagnosis was performed at the Institute of Pathology. The samples were processed according to gold standards, including IHC, to definitively confirm the diagnosis. Specimens obtained by SBMA were smeared, air-dried, stained using the May-Grunwald and Giemsa technique, and evaluated by light microscopy in the Hematology Laboratory for Cytomorphologic Diagnostics. After evaluation of cytomorphology in those specimens that were identified as hematological malignancy, the molecular techniques of flow cytometry, FiSH, and cytogenetic analysis were performed according to international standard procedures.

Statistical analysis was performed using the current software package from IBM SPSS 27.0 (IBM Corp., Armonk, NY, USA). For statistical analysis of the obtained data, we calculated the percentages of agreement of the histological and cytological diagnoses as well as their sensitivity and specificity. To measure the agreement between histology and cytology, we calculated Cohen’s kappa coefficient. The significance level is <0.05.

## 3. Results

Overall, 44 patients who underwent simultaneous histologic and cytologic evaluation of a spinal lesion suspected of malignancy were included. The mean age was 66 (±12) years and the majority were men (61%). All patients underwent cytopathologic specimen collection in addition to histopathology, as described above. In two cases (5%), no intralesional material for cytology could be obtained by SBMA. Thus, cytological evaluation was performed in 42 cases (95%). The collection of additional samples for cytology specimens was safe with no significant extension of surgery time. No complication was observed due to the additional diagnostic procedures. Figure 1 shows examples of cytological light microscopic images obtained by bone marrow aspiration from vertebral body lesions suspected of malignancy (Figure 1).

In total, 39 patients (89%) were diagnosed with malignant disease in the histopathological evaluation, 28 (72%) of which proved to have solid malignancies and 11 (28%) of which proved to have hematologic malignancies in the sense of multiple myeloma (Table 1). The subtypes of solid malignancies are shown in Table 2. All detected solid tumors corresponded to a primary tumor from another organ system. No primary malignant tumors of the spine were identified.

In the cytologic evaluation, malignant disease was detected in 37 cases (88%), of which 8 (21%) could be identified as multiple myeloma and 29 (76%) as solid malignant tumors (Table 1).

In 40 of 42 cases where cytological examination was performed successfully, cytological and histological diagnoses were consistent. This resulted in an agreement of 95.2% with a Cohen’s kappa of 0.77, corresponding to a strong agreement. The cytological diagnosis thus had a sensitivity of 0.97 and specificity of 0.80 concerning the identification of malignant processes. The diagnostic certainty of cytology, for the identification of a malignant process, is 95% (Table 3).

Furthermore, differentiation of the specimen into benign lesions, hematological neoplasm, and solid malignancies also proved coincident in 95% of cases (Cohen’s kappa of 0.902), corresponding to almost complete agreement (Table 4).

## 4. Discussion

In this study, the additional cytologic evaluation of spinal lesions with suspected malignancies using SBMA during emergency or highly urgent spinal surgery was analyzed for the first time in terms of its procedural safety, sensitivity, specificity, and diagnostic certainty in a collective of 44 patients. Histopathologic diagnosis as the clinical gold standard was used as the reference. In 39 patients (89%) with suspected new-onset malignant spinal lesions, histologic examination confirmed a malignant disease: 28 cases (72%) were identified as solid malignant tumors and 11 (28%) as hematologic neoplasm corresponding to multiple myeloma. In patients with myeloma, the standard-of-care diagnostic could be performed from additional obtained aspiration for assessment of prognostic markers including FiSH analysis. Within solid tumors, metastases from breast carcinoma were most commonly identified, followed by prostate carcinoma, small cell neuroendocrine tumors, bronchopulmonary adenocarcinoma, and hepatocellular carcinoma. This distribution of the various tumor entities is consistent with the current literature [1].

In the current study, we were able to show that in 95% of cases, sufficient additional cytological diagnostics can be performed through SBMA from the vertebra suspected of malignancy without posing additional risk to the patient. In two cases, no meaningful cytological sample could be obtained from the vertebral body in the described cohort. Both cases were vertebral bodies affected by multiple myeloma with a high percentage of plasma cells. It is conceivable that the high proportion of plasma cells in the bone marrow of the vertebral bodies made the marrow so viscous that it could not be aspirated and therefore no meaningful samples could be obtained. In both cases, however, additional bone marrow aspirates were obtained from the iliac crest. In these samples, multiple myeloma could be cytologically detected in both cases, which was confirmed by the histopathological results from the vertebral bodies in the course.

Cytologic diagnosis from spinal lesions by percutaneous FNAB has been frequently described and studied. FNAB is performed preoperatively in elective cases and radiologically supported through sonography [19], fluoroscopy [15], computed tomography (CT) [16], or magnetic resonance imaging (MRI) [17]. According to current findings, a diagnostic certainty of 70–93% can be achieved by FNAB [16,19,20,21,22]. Overall, this technique appears to be slightly inferior to histological core needle biopsy (CNB), which is considered the gold standard of percutaneous minimally invasive procedures, having a diagnostic certainty of 90–98% [23,24,25]. However, the diagnostic certainty of the intraoperative cytologic analysis of SBMA during spinal surgery has not yet been described or evaluated. In routine clinical practice, cytologic examination of spinal lesions by FNAB is performed independent of spine surgery and as part of the elective program. The patients who benefit from the intraoperative cytologic analysis of SBMA presented in the current study were those who required emergency or very urgent spinal surgery to avoid neurologic deficits and immobilization. For these patients, elective radiologically guided FNAB is often not an option. The technique presented in the current study makes it possible to provide this group of patients with cytological diagnostics with a high degree of diagnostic certainty, expanding the diagnostic spectrum and increasing patient safety.

Regarding malignancies, our data showed a 95% compliance of successfully obtained SBMA and derived histopathology (Cohen’s kappa: 0.773). Sensitivity and specificity were calculated to be 97% and 80%, respectively. With a diagnostic safety of 95%, SBMA shows high results comparable to those of elective radiologically guided FNAB [15,16,17]. For the cytological evaluation of bone marrow obtained during surgery from a spinal lesion suspected of malignancy, these parameters have not yet been described in the literature. Since most patients with metastatic spine disease undergo mostly urgent intervention with little time for prior oncologic preparation, the high SBMA scores can be regarded as especially meaningful for both the interdisciplinary team and the patient, potentially reducing further painful and stressful diagnostic steps.

In 1988, Findlay et al. described the smear preparation of bone cylinders derived via CNB during spinal surgery. They cytologically assessed 35 cases, reporting successful cytologic diagnosis in 93% of cases, with 97% concordance between cytologic and histologic diagnoses. However, in this study, cytology was not further evaluated for sensitivity and specificity. In addition, Findlay et al. described that solid malignant processes could be identified as such on cytology and that myeloma diagnosed in two cases within the cohort could already be diagnosed with certainty on cytology [26]. However, these correlations were not statistically described in the paper.

In the current study, we were able to statistically demonstrate the ability to differentiate benign lesions, hematologic neoplasms, and solid malignancies using SBMA in a collective of 44 patients, for the first time. Regarding these three clinically highly relevant groups, cytology showed a concomitance of 95% with histology (Cohen’s kappa of 0.902). This corresponds to an almost complete agreement between the histology and cytology results, urging further implementation of this highly feasible technique as a routine clinical diagnostic tool. Due to the characteristics of cytological diagnosis, it is faster and easier to perform than histological diagnostics and, therefore, more favorable.

Even without regarding specific subentities, the diagnostic and therapeutic pathways of patients vary greatly depending on the three entities: solid neoplasms, hemic neoplasms, or benign lesions. For the treatment team, but even more for the individual patient, it is of utmost urgency to identify those future steps, relieving patients’ anxiety and developing individual treatment plans. Thus, the performance of additional cytological diagnostics may be of the highest importance in terms of health economics, the safety of the individual patient, time to diagnosis confirmation, and the initiation of further diagnostic and therapeutic steps. Currently, in clinical routine, a tissue sample is obtained intraoperatively from patients who are operated on for a spinal metastasis, which is then examined histopathologically. In the case of the mostly bony samples, up to 14 days pass before a final diagnosis can be made due to the complex processing of the tissue. In the current study, we showed that malignant lesions can be detected with a high sensitivity and specificity by spinal aspiration cytology and assigned to specific subgroups. Based on the current study, future studies should determine if the time until confirmation of the diagnosis and initiation of a specific therapy can be shortened by the additional intraoperative SBMA and its cytological evaluation and if immunocytochemical analysis of detected solid tumors may deliver a definitive diagnosis.

Furthermore, intraoperative SBMA cytology as described by us can be performed as part of a rapid on-site evaluation (ROSE) [27]. This can provide a simple and cost-effective alternative to intraoperative cryosection procedures for rapid diagnosis, for which a diagnostic certainty of 89% was described [28]. Intraoperative ROSE of spinal lesions suspected of malignancy might offer many benefits in clinical practice. With the technique presented in the current study, it is possible for any spine surgeon to obtain SBMA biopsies for cytological evaluation during surgery. However, the samples must be evaluated by hemato-oncologists or cytopathologists, who are not ubiquitously available. This availability is crucial as it is needed for emergency diagnostics. In the context of ROSE, however, it is possible to create, stain, and digitize cytological preparations within a few minutes by means of automatic staining machines and scanners, which can be established in the operating area. These can then be used for telepathological evaluation. The costs of establishing such an infrastructure are far below the costs that would be required for a hemato-oncology or cytopathology service ubiquitously and always available. Small hospitals in the standard care sector would especially benefit from this. In addition, the procedure presented in the current study can be used within the framework of ROSE for timely, reliable, and cost-efficient screening of malignant processes, for example, in osteoporotic fractures, which cannot always be reliably clinicoradiologically distinguished from malignant processes. Further studies should determine if obtaining cytological samples during an emergency or highly urgent spinal surgery via SBMA, as presented by us, may be utilized in ROSE.

## 5. Conclusions

The current study is the first to demonstrate the highly relevant value of supplemental intraoperative SBMA cytology from spinal lesions. SBMA cytology is a highly feasible, sensitive, and specific tool for the identification of malignant lesions. Furthermore, this method can be used to efficiently differentiate between solid and hematologic neoplasms. With a high diagnostic certainty, SBMA extends the diagnostic spectrum with a time and cost benefit possibly improving patients’ care and safety.

## Figures and Tables

**Figure 1 diagnostics-12-00310-f001:**
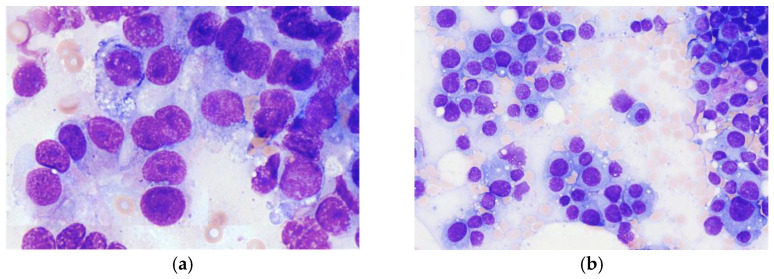
Example light microscopy images of cytologic specimens obtained by aspiration from vertebral body lesions suspected of malignancy: (**a**) Smear from aspiration cytology of bone marrow from a malignant spinal osteolytic lesion of unknown origin. The image shows accumulations of large atypical nonhematologic cells with immature nuclei and basophilic cytoplasm with sparse vacuolization consistent with adenocarcinoma. The patient had a PSA level of 4300 μg/L and final histology revealed prostate carcinoma; 100× magnification. (**b**) Smear from aspiration cytology from an osteolytic lesion of the spine. The specimen shows subtotal infiltration with mature but atypical plasma cells of polymorphic size, including giant cells corresponding to plasma cell myeloma; 40× magnification.

**Table 1 diagnostics-12-00310-t001:** Number of benign, hematologic malignant, and solid malignant lesions identified by histology and cytology.

Lesion Type	Histology	Cytology ^1^
Benign	5	5
Hematologic malignant	11	8
Solid malignant	28	29
Total malignant	39	37
Overall total	44	42 ^1^

^1^ In 2 of the 39 cases included in the study, no cytological evaluation could be performed on the samples obtained.

**Table 2 diagnostics-12-00310-t002:** Number of solid malignant lesion subtypes identified in histology ^1^.

Malignant Lesion Subtype	Number
Breast carcinoma	8
Prostate carcinoma	4
Small cell neuroendocrine carcinoma	3
Hepatocellular carcinoma	3
Bronchopulmonary adenocarcinoma	3
Malignant melanoma	2
Renal cell carcinoma	1
Biliopancreatic adenocarcinoma	1
Epithelioid angiosarcoma	1
Bronchopulmonary squamous cell carcinoma	1
Adenocarcinoma of unknown origin	1

^1^ All detected malignant lesions corresponded to a metastasis of an extraspinally localized primary tumor. There were no primary malignant spinal tumors in the investigated population.

**Table 3 diagnostics-12-00310-t003:** Sensitivity, specificity, and diagnostic certainty of cytology from the target vertebra to identify a malignant lesion in relation to the diagnosis of the reference pathology.

	Cytology from the Target Vertebra
Sensitivity	0.97
Specificity	0.80
Diagnostic certainty ^1^	0.90

^1^ Diagnostic certainty = sum of correct findings (malignant or benign)/number of subsampled cases.

**Table 4 diagnostics-12-00310-t004:** Number of matches between cytological and histological diagnoses.

	Histology	Cytology	Number of Matches ^1^
Benign	5	5	4
Hematologic malignant	11	8	8
Solid malignant	28	29	28

^1^ Number of matches = cases identified as benign, hematologic malignancy, or solid malignancy by both histology and cytology.

## Data Availability

The data presented in this study are available within this article. All data generated or analyzed during this study are included in this published article. A summary of the data can also be provided upon request.

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
