# Peer review of "Surgical Site Cytology to Diagnose Spinal Lesions"

_diagnostics, 2022, doi:10.3390/diagnostics12020310_

Round 1

Reviewer 1 Report

The authors describe diagnostic accuracy of cytology in evaluation of spinal lesions. The study is well-planned but lacks in many aspects. My comments are as follows:

  1. Utility of cytology in evaluation of spinal lesions is well established. The authors need to establish what is novel about their study.
  2. The cases included metastases from a wide range of organ systems. Was any ancillary technique like immunocytochemistry used for confirmation of the primary? Was any ancillary technique like molecular techniques done using the cytology material to aid in patient management?
  3. It would have been more interesting if the authors would have studied cytology of some primary spinal lesions too, especially in cases which are a diagnostic dilemma clinico-radiologically. 
  4. Multiple grammatical mistakes throughout the manuscript need to be taken care of.
  5. The title is too lengthy and difficult to understand.

Reviewer 2 Report

General comment: The authors presented an interesting and original work concerning to the role of intraoperative cytology in the management of spinal neoplasms.

The manuscript is written in a comprehensive way. Despite this, it should be revised by an English native speaker. E.g. “This results in an…” (line 146)

Title: The title is adequate.

The keywords should be different from those used in the title.

Abstract: It is adequate.

Introduction: It is adequate. The authors provided an adequate overview of the thematic.

Materials and methods: Please provide the full name before the abbreviation STROBE.

Results: They are clearly presented and supported by the Figures and Tables.

Discussion: Please provide the full name before the abbreviation CT and MRI.

Conclusion: The conclusion is based on the results.

Recommendation: The manuscript should be accepted for publication after a Minor revision.

Round 2

Reviewer 1 Report

Following the first round of review, the authors have edited the manuscript well, bringing out the significance of their study in a better way. They have also satisfactorily provided point wise clarification to all the comments.

Only one minor typographical error needs to be taken care of:

'Spinal manifestations of a primary tumor located in another organ system is by far the largest group, by far.': 'By far' is being repeated twice.

Author Response

We thank you again for carefully reading our manuscript. We adopted the requested changes to the manuscript. The change was tracked within the manuscript. In the following we provide a response regarding your concern.

Point 1: Only one minor typographical error needs to be taken care of: 'Spinal manifestations of a primary tumor located in another organ system is by far the largest group, by far.': 'By far' is being repeated twice.

Response 1: Thank you for carefully reading our manuscript. As you suggested we corrected our mistake. Please refer to line 36.